# OpenReview forum: "CyberGym-E2E: Scalable Real-World Benchmark for AI Agents' End-to-End Cybersecurity Capabilities"
_ICML.cc/2026/Conference — ICML 2026 regular_

### Official Review · Reviewer_LR3s · 2026-03-12

**Soundness:** 3
**Presentation:** 3
**Significance:** 1
**Originality:** 1
**Overall Recommendation:** 2
**Confidence:** 4

**Summary:**

The paper introduces a tool for benchmarking AI agents in the area of cybersecurity. By filtering the vulnerability dataset OSS-Fuzz, they create a benchmark suite containing 120 open source projects and implement a pipeline for testing AI agents across different harnesses, namely Claude Code, Codex, Gemini CLI and OpenHands. The large language models are evaluated in the patch-only setting, and the end-to-end setting, where vulnerabilities are discovered, proof-of-concept inputs are generated and tested, and a patch is generated. The performance of different configurations is then compared.

**Compliance With Llm Reviewing Policy:**

Affirmed.

**Key Questions For Authors:**

1.	In table 3, the S4 value shows that the agent often identifies and patches a different vulnerability. If the method is run again with the first vulnerability fixed, do the agents then succeed in finding vulnerability? I.e. do the agents only get distracted by another vulnerability, or do they not find the selected vulnerability at all?
2.	Do the agentic model configurations show similar performance on other benchmarks?
3.	In table 3 in the end-to-end setting: If we only look at cases that pass S1 and then calculate the percentage that also pass S3 by dividing the S3 result by S1, we obtain the following patch success rates: (77.1, 58.6, 58.1, 68.5, 76.4). Why do they notably differ from the patch-only setting?

**Limitations:**

Yes

**Strengths And Weaknesses:**

*Soundness:* The benchmark builds on the well-known dataset OSS-Fuzz to determine the capabilities of different agents. However, there are some shortcomings: The paper would benefit from additional scientific context and the related work could be better compared to. For example, in the introduction in line 22, it is stated that the benchmarks “suffer from other issues”, which should be explained more clearly. Also, according to line 69 and 70, BountyBench and SeCodePLT “do not provide a realistic environment for ai agentic evaluation”. Such claims need to be grounded.

Additionally, important details are missing to fully understand the benchmark, for example how the migration to a newer OS is handled and how it is made sure that no new vulnerabilities are introduced because of this, as it could skew the results.
Finally, the paper could benefit from a more thorough analysis and discussion of the results. For example, in table 3 in the end-to-end setting: If we only look at cases that pass S1 and then calculate the percentage that also pass S3 by dividing the S3 result by S1, we obtain the following patch success rates: (77.1, 58.6, 58.1, 68.5, 76.4), which differ quite a bit from the patch-only setting. It’s not quite clear why there is such a difference, especially as the significance of the results is not shown.

Also in 4.4: Time and cost budget, it is said that further increasing the budget over $5 provides diminishing improvement. While the relative jump does decrease, there are definitely notable gains for some models.
:

*Presentation:* The structure of the paper is coherent and it has good grammar and spelling. Figures illustrate the content in a good manner, apart from some minor details. For example in Figure 2, “S1 PoC triggers vuln” should probably be between “PoC and crash logs” and “Patch generation”. Figure 3 could benefit from further explanation of the different kinds of steps (Agent, Environment, ..).


The explanation of the method is a bit shallow and quite a bit of information is repeated 3 or more times. Thus, the space of the paper could be used more efficiently, and more details could be provided.

Additionally, there are some minor issues, like in line 358 (“doubling the budget from $1 to $5”).


*Significance:* The area of cybersecurity is highly relevant and to safely employ AI agents for such purposes, its capabilities need to be tested. Benchmarking these models thus benefits the community. However, there are no major new insights, and the contribution is very limited. An existing, popular dataset (OSS-Fuzz) is used and filtered, and existing AI models and harnesses are employed. While the tool might be useful for the community, it does not sufficiently advance understanding.

*Originality:* In terms of content the paper is very close to its related work with very limited originality, differing mainly in the dataset and through the introduction of the “end-to-end” evaluation. Additionally, the benchmark is not sufficiently compared to the related work. In the experiments, different models and harnesses are evaluated. However, it would be interesting to see whether the same configurations show different results in related benchmarks.

---

> ### Author Rebuttal · Authors · 2026-03-31
>
> We thank the reviewer for the constructive feedback.
>
> **Target vulnerability discovery (Q1):**
> Both situations exist: in some cases, agents would find the ground-truth one if the distractor were removed; in other cases, they do not find the intended vulnerability at all. We conducted additional experiments on the tasks that passed S3 but not S4, re-running the agent after applying the patch for the first vulnerability. About 10% of them subsequently found and patched the ground-truth vulnerability.
>
> **Agent performance on other benchmarks (Q2):**
> We appreciate this question. We note that CyberCycle is a benchmark rather than a new agent harness, and the current evaluation already costs over $30k in API fees, so running the same configurations on external benchmarks was not our primary focus and not feasible within our budget. However, we collected comparable data from public sources to provide some context.
> According to the Anthropic system cards, Claude Opus 4.5 achieves 50.6% on CyberGym and 82% on Cybench, while Claude Sonnet 4.5 achieves 28.9% on CyberGym and 60% on Cybench.
>
> **Patching performance gap (Q3):**
> The difference might be because the two settings place different demands on the agent's task and attention. In the patch-only setting, the agent’s entire context is focused on patch generation. In the end-to-end setting, the agent performs all three stages sequentially, accumulating context from earlier exploration. When the agent reaches the patching phase, its context window contains extensive exploration history, which may dilute focus and reduce patching effectiveness. It suggests that multi-step cybersecurity workflows impose a cognitive overhead on agents beyond the difficulty of each individual subtask.
>
> **Soundness (related work):**
> We will revise the paper to make the claims clearer. We include a detailed comparison in Table 1 and discuss the specifics in Section 2.1. For example, at line 101, we note that BountyBench also evaluates end-to-end agent capabilities on real-world vulnerabilities across their lifecycle, but suffers from scale limitations, covering only 40 tasks, as it relies entirely on manual task curation, and SeCodePLT constructs tasks from synthetic modifications of real code rather than evaluating agents on the entire repository.
>
> **Soundness (OS migration):**
> We migrate the entire codebase of the project and keep the compilation scripts the same, and re-validate that the ground-truth vulnerability still exists. If migrating old code to a new OS introduces additional vulnerabilities, these would appear as different vulnerabilities in S3, while the ground-truth vulnerability is validated separately in S4. Although we did not observe such a situation, we note that if an agent discovers additional vulnerabilities arising from running older code on a newer OS, this would be a useful and interesting finding, making the benchmark more valuable rather than less reliable. The original benchmark integrity is not affected.
>
> **Soundness (diminishing claims in 4.4):**
> We agree that the phrasing could be more precise. While the relative gains decrease at higher budgets, there are still meaningful absolute improvements for some models (e.g., Opus 4.5 improves from 11.0% to 19.2% between 5 dollar and 10 dollar). We will revise the text to more accurately characterize the cost-performance tradeoff.
>
> **Presentation:**
> Thanks for your detailed suggestions. We will incorporate them and fix typos.
>
> **Significance and Originality:**
> We respectfully disagree with the assessment that the contribution is limited. We would like to highlight several aspects:
>
> 1. **The benchmark provides a high-quality, validated dataset that goes substantially beyond OSS-Fuzz.** Although the vulnerability data is sourced from OSS-Fuzz, the raw OSS-Fuzz dataset does not provide reproducible environments compatible with modern agent frameworks, and does not include infrastructure for running unit tests needed to validate that a patch does not break the normal functionality of the project. CyberCycle constructs all of these missing components: unit tests, dockerized agent-compatible environments, ground-truth patches linked to specific commits, and a complete evaluation pipeline. Constructing and validating these artifacts at scale is a non-trivial contribution.
>
> 2. **CyberCycle is the largest end-to-end cybersecurity benchmark.** As shown in Table 1, BountyBench, the closest prior work in end-to-end evaluation, contains only 40 tasks. CyberCycle provides 615 tasks (now grown to 910 across 138 projects), enabling statistically meaningful comparisons across models and harnesses that were not previously possible.
>
> 3. **The evaluation yields novel insights.** These include the impact of harness architecture on performance (Section 4.3), meaningful gains from cross-run feedback (Table 6), and the identification of adversarial agent behavior such as capability misrepresentation and selective reporting (Section 4.5).

---

> > ### Author Rebuttal · Reviewer_LR3s · 2026-04-03
> >
> > Answer to Q1: In my opinion this implies that in the majority of cases, the agent is not distracted by another vulnerability, but fails to find it.
> >
> > Answer to Q2: The cost restriction is understandable.
> >
> > Answer to Q3: My question is partially answered. What if the agent’s context is cleared and the patch scenario is run?
> >
> > Regarding Significance and Originality: I understand that the implementation took a significant amount of time, but the conceptional contribution is still severely limited.

---

> > > ### Author Response · Authors · 2026-04-05
> > >
> > > We thank the reviewer for the feedback and for the continued engagement.
> > >
> > > **Q1:** We agree with the reviewer's interpretation: in the majority of cases, the agent fails to find the intended vulnerability. This is also meaningful evidence that shows CyberCycle is valuable for the community to continue to identify gaps and improve agents' cybersecurity capabilities.
> > >
> > > **Q3:** This is exactly the patch-only setting evaluated in Table 3: the agent receives the PoC and crash logs with a fresh context and performs only the patching task. It achieves a higher success rate, as shown in Table 3.
> > >
> > > **Significance and Originality:** We thank the reviewer for acknowledging the significant implementation effort. We respectfully maintain that the contribution is meaningful. Widely adopted benchmarks such as SWE-bench are valued by the community because they provide carefully constructed, large-scale evaluation frameworks. CyberCycle serves a similar role for cybersecurity: it is significantly larger than the closest end-to-end benchmark (BountyBench, 40 tasks) and the evaluation has yielded novel findings, including the distinction between distraction and inability identified in Q1, the impact of harness architecture on performance (Section 4.3), meaningful gains from cross-run feedback (Table 6), and the identification of adversarial agent behavior such as capability misrepresentation and selective reporting (Section 4.5).
> > > Besides the benchmark dataset itself, the proposed agent-assisted pipeline is also a conceptual contribution, demonstrating the practice of leveraging agents to facilitate cybersecurity dataset construction.
> > > We believe high-quality benchmark infrastructure is a valuable research contribution and is meaningful to the community.

---

### Official Review · Reviewer_xPYm · 2026-03-12

**Soundness:** 3
**Presentation:** 3
**Significance:** 4
**Originality:** 3
**Overall Recommendation:** 5
**Confidence:** 2

**Summary:**

This paper introduces a benchmark named CyberCycle to evaluate the end-to-end capabilities of AI Agents in several cybersecurity tasks that span multiple stages of the real-world software vulnerability lifecycle. Unlike prior benchmarks that focus on static analysis, CyberCycle evaluates agents in a repo-level, dynamic, containerized environment, where they must compile code, run sanitizers, interpret execution logs, and operate in realistic software engineering settings. A central contribution of the paper is its automated benchmark construction pipeline, which recovers vulnerable versions, PoCs, fixing commits, and validation environments from historical OSS-Fuzz vulnerabilities.

**Compliance With Llm Reviewing Policy:**

Affirmed.

**Key Questions For Authors:**

I don't have many questions about this paper; just think it focuses solely on evaluations related to code vulnerabilities, and its claim of addressing "cybersecurity" is a slight overstatement.

**Limitations:**

yes

**Strengths And Weaknesses:**

Soundness

Strengths: The highlight of this paper is its pipeline of a relatively systematic automated benchmark. Compared to previous high-quality benchmarks that are costly to scale, this work attempts to automatically recover vulnerable versions, PoCs, fix commits, and verification environments from real-world historical vulnerabilities. Furthermore, the paper adopts a dynamic, containerized, repo-level evaluation environment rather than function-level or static code snippet settings, making the benchmark more closely aligned with real-world software engineering and vulnerability handling processes. Based on these two points, This approach to constructing agent benchmarks holds great value.

Weaknesses: However, the title and abstract use the broad term "cybersecurity," but the current benchmark's coverage is actually quite specific. It primarily originates from OSS-Fuzz and focuses on memory-safety vulnerabilities. This significantly differs from web security, privilege escalation, logic flaws, and ecosystem issues in high-level languages within the broader scope of cybersecurity. Therefore, the current phrasing could easily lead readers to overestimate the benchmark's applicability.

Presentation

This paper is clearly structured and easy to understand.

Significance

The significance of this paper is clear. Current evaluations of AI agents' security capabilities often remain at the level of static code analysis, CTF-style tasks, or patch-only settings, making it difficult to truly reflect agent performance in complex software engineering environments.

Originality

First, the authors propose a relatively complete automated benchmark construction pipeline, attempting to break through the bottleneck of scaling high-fidelity security benchmarks. Second, the paper advances the evaluation paradigm from function-level or static code analysis to repo-level, dynamic containerized environments, and employs S1-S4 metrics to provide a fine-grained characterization of agent capabilities.

---

> ### Author Rebuttal · Authors · 2026-03-31
>
> We thank the reviewer for the constructive feedback.
>
> Following the reviewer’s suggestion, we will add a dedicated Limitations and Scope section that explicitly discusses (1) the dataset's current focus on memory-safety vulnerabilities, (2) the characteristics inherited from OSS-Fuzz as the primary data source, and (3) the gap between this scope and "cybersecurity" broadly. We will also note these limitations in other sections where we discuss the scope of the benchmark.

---

### Official Review · Reviewer_VX5Z · 2026-03-12

**Soundness:** 3
**Presentation:** 3
**Significance:** 2
**Originality:** 2
**Overall Recommendation:** 4
**Confidence:** 4

**Summary:**

The paper introduces a benchmark to evaluate the capabilities of AI agents to identify and repair code vulnerabilities. The paper also presents a methodology to construct the benchmark, which ends up featuring 615 real-world vulnerabilities in 120 open-source projects. Evaluation covers both vulnerability discovery and patch generation and provides insights into the comparative performance of underlying LLMs and the performance of agentic harnesses.

**Compliance With Llm Reviewing Policy:**

Affirmed.

**Ethical Review Concerns:**

Paper does not discuss the ethical implications of improving agentic capabilities to discover vulnerabilities and create exploits.

**Ethical Review Flag:**

Flag this paper for an ethics review.

**Ethics Expertise Needed:**

["Privacy and Security (e.g., personally identifiable information)"]

**Key Questions For Authors:**

1. How reproducible are the Dockerfiles? Do you plan to provide the container images as part of the benchmark? Or instructions/scripts to build historical (Ubuntu 16.04) environments?

2. Do the results in Table 1 for S3 also include patches for a different vulnerability than the ground truth? It seems that validation of correct vuln patch is done only on S4.

**Limitations:**

The paper must acknowledge the negative impacts of accelerated vulnerability discovery.

**Strengths And Weaknesses:**

**Strengths**
+ Semi-automated benchmark construction
+ Benchmark stages allow for the evaluation of underlying LLMs and the evaluation of agentic harnesses

**Weaknesses**
- The harness comparison is limited
- The benchmark construction does not seem scalable, since Step 4 is manual
- The benchmark construction is not evaluated
- No explanation how $10 cap was selected
- Unclear statistical significance of the evaluation results -- one run only?
- Unclear token budget is a useful termination criterion, as opposed to context exhaustion; this muddles the comparison of model capabilities

*Harness comparison*. Section 4.3 looks at only two elements of the agentic harness, file strategy and task tracking. It would be interesting to broaden the comparison with more dimensions (granularity of chosen tools, parallel tool execution, etc.) and also to gather more concrete measurements of the impact of these elements on the benchmarked performance.

*Benchmark construction scalability*. Step 1-3 of the benchmark construction pipeline seem to be automated, which Step 4 relies on human review. It is unclear how this results in a scalable pipeline, one of the stated requirements.

*Benchmark construction quality*. The process of building the benchmark is not evaluated on how well it automates the task of selecting and documenting POCs. At a minimum it'd be useful to report how much was filtered out through the manual review.

*Budget cap*. It seems that all of the models improve given more time and more token budget. The paper should explain how the $10 and 90-minute caps were chosen. It'd be useful to plot cost-per-vulnerability distribution curves for each models x harness combo.

*Statistical significance*. Given variability/non-determinism in model response, the benchmark run should be repeated a few times to ensure the reported success rates are indeed representative. It is not clear from the paper whether results in Table 3 were from a single run through the benchmark, or averaged across multiple runs.

---

> ### Author Rebuttal · Authors · 2026-03-31
>
> We thank the reviewer for the constructive feedback.
>
> **Harness comparison (W1):**
> We have conducted additional analysis of agent trajectories and provide a more detailed comparison below.
>
> Tool granularity: Codex and OpenHands are both bash-centric, routing most actions (cat, sed, rg, head) through shell commands. In contrast, Claude Code and Gemini CLI use more granular toolsets including dedicated tools for reading, searching, and editing files.
>
> Parallel tool execution: We did observe that agents support parallel execution. However, the parallelism is mostly limited to reading multiple files or grepping across files simultaneously. Because the agent needs to retrieve content and analyze it before deciding the next step, the main task progression remains inherently sequential and cannot be meaningfully parallelized.
>
> We also provide qualitative analysis of agent behavior patterns in Section 4.5, including exploration strategies, failure modes, and adversarial behavior, which offers additional insight into how harness design affects performance.
>
>
> **Benchmark construction scalability (W2):**
> Step 4 (human validation) is intentionally retained to ensure high data quality, but it is a validation step, not a construction step. The automated pipeline (Steps 1-3) handles the heavy lifting of identifying patches, preparing build environments, and extracting test suites.
> In Stage 4, the acceptance rate is 74%, meaning the majority of pipeline-generated candidates pass expert review.
>
> Since submission, the pipeline has continued to run on more recently released vulnerabilities and scaled the dataset from 615 to 910 tasks across 138 projects, demonstrating scalability in practice.
>
> **Benchmark construction quality (W3):**
> We provide the filtering criteria and statistics following the stages illustrated in Figure 1:
>
> Stage 1: This stage filtered out approximately half of the data, leaving about 1,400 candidates. Vulnerabilities were excluded if the patch commit message was uninformative or spanned unrelated issues.
>
> Stage 2: About 15% of the data failed this stage (e.g., could not find the vulnerable commit, PoC did not behave as expected). About 1,200 candidates remained.
>
> Stage 3: After agent-assisted test identification, about 800 candidates remained. Failures at this stage were primarily due to the agent being unable to resolve compilation issues, failing to identify tests, or making excessively invasive changes to build scripts.
>
> Stage 4: 74% passed, and 26% were rejected, yielding the final 615 tasks. At this stage, the human expert reviewed each entry for: (1) obscure or redundant code and comments, (2) excessive on-the-fly code changes that break idempotency, (3) redundant dependencies, (4) base image, (5) project metadata, and (6) unit tests.
>
> **Budget cap (W4):**
> For Sonnet 4.5, the success rate already converges at around $5 as shown in Table 5. For Opus 4.5, the per-token price is substantially higher, so we added an additional no-cap evaluation. For GPT-5.2-Codex and Gemini 3 Pro, we plotted cost vs. pass-rate curves (https://imgur.com/BDCqjQc https://imgur.com/2UFBEIt) and observed that success rates also converge near $10.
>
> From this analysis, $10 is a reasonable budget cap that captures most of the achievable performance for the majority of models.
>
> **Statistical significance (W5):**
> The results in Table 3 are from single runs. The current evaluation in the paper already costs over $30k in API fees, making multiple runs prohibitively expensive. This is a common constraint in large-scale agent benchmarks: prior work such as SEC-bench, SecRepoBench, and CyberGym also report single-run results likely due to similar resource limitations.
>
> **Context exhaustion (W6):**
> In practice, we did not observe much context exhaustion in our evaluation. Because the agent must carry the entire previous context at each step, total token consumption grows cumulatively. This compounding cost means agents typically exhaust their cost budget well before hitting the context window limit. We also justified the choice of $10 in W4.
>
> **Docker image (Q1):**
> Yes, we will release the container images as part of the benchmark.
>
> **Stage 3 result (Q2):**
> Yes. S3 validation includes the different vulnerability found by agent in S1. The validation of the patch on the ground-truth vulnerability is done by S4.
>
> **Ethics:**
> We will add more details to the impact statement addressing dual-use risks. We also note that this is precisely why CyberCycle evaluates the full lifecycle including the defensive capability (patch stage), rather than focusing solely on offensive capabilities. By benchmarking both discovery and remediation together, we aim to support the development of AI systems that strengthen defense, not just accelerate attack capabilities.

---

> > ### Author Rebuttal · Reviewer_VX5Z · 2026-04-03
> >
> > For the benchmark construction scalability, I am not sure what the distinction is between "validation step" and "construction step", with the former being apparently optional. If a benchmark contains unvalidated samples, it is not a good benchmark.
> >
> > For the budget cap, the new provided chart does not indicate that success rate converges near $10, since the x-axis does not continue past $10.

---

> > > ### Author Response · Authors · 2026-04-05
> > >
> > > We thank the reviewer for the feedback and for the continued engagement.
> > >
> > > **Benchmark construction scalability:** We apologize for the confusion in our previous response.
> > >
> > > In our pipeline (Figure 1), the construction steps are Steps 1–3 (identifying patches, preparing build environments, and extracting test suites), and the validation step is Step 4 where a human expert reviews every task.
> > >
> > > We agree with the reviewer that all tasks need to be validated. This is exactly why Step 4 exists: all tasks in the final dataset have been validated by human experts, and this validation step is mandatory, not optional. We only keep the tasks that pass expert review.
> > >
> > > The agent-enhanced Steps 1–3 reduce the human effort from manually constructing each task to only validating the agent's output. From a scalability perspective, we acknowledge that the pipeline is not fully automated and still requires human validation to ensure high quality, but the agent-assisted construction substantially facilitates the process.
> > >
> > >
> > > **Budget cap:** We thank the reviewer for the feedback on the cost curves. The curves show some diminishing returns before \\$10, and we also acknowledge that \\$10 does not represent full convergence for all models. The \\$10 cap was chosen to enable fair cross-model comparison and control the budget, as the current evaluation already costs over \\$30k in API fees.
> > >
> > > Also, the budget cap is an evaluation parameter and does not affect the benchmark dataset itself. Model providers or other researchers with more resources who wish to evaluate with a higher budget can directly reuse our evaluation pipeline and dataset. We will revise the paper to more accurately characterize this point.

---

### Official Review · Reviewer_kzxh · 2026-03-12

**Soundness:** 3
**Presentation:** 3
**Significance:** 3
**Originality:** 2
**Overall Recommendation:** 4
**Confidence:** 4

**Summary:**

The paper introduces CyberCycle, a benchmark for evaluating AI agents
on end-to-end cybersecurity tasks including vulnerability discovery,
PoC generation, and patch generation. The benchmark is built from 615
real vulnerabilities across 120 open source projects and makes it
possible to to validate whether generated PoCs trigger crashes and
whether patches fix the bugs without breaking
functionality. Experiments with several modern AI agents show that
while models perform well at patch generation when given crash
information, they struggle in the end-to-end setting where
vulnerabilities must be discovered and exploited before being patched.

**Compliance With Llm Reviewing Policy:**

Affirmed.

**Final Justification:**

Discussed in response to rebuttal.

**Key Questions For Authors:**

1. The benchmark relies on OSS-Fuzz vulnerabilities and programs that
process structured inputs. How representative is this dataset of the
broader landscape of real-world vulnerabilities and software systems?
How easily can this benchmark be extended to encompass new vulnerability
types?

2. Section 3.3 mentions manual review and exclusion of certain
projects or vulnerabilities. Is it possible to provide more details
about the exclusion criteria and statistics on how many instances were
removed at each stage?

3. Are there any insights on how future work could improve AI agent
frameworks for vulnerability discovery beyond simply improving the underlying models? For example, are there observations
from the evaluation about agent design, tool usage, or interaction
strategies that could guide the development of more effective security
agents?

**Limitations:**

The paper discusses some limitations, such as the focus on C and C++
programs and the reliance on OSS-Fuzz data. However, several
limitations are not sufficiently addressed. In particular, the
diversity of programs and vulnerability types is limited due to the
dependence on OSS-Fuzz, and the benchmark mainly captures
crash-triggering vulnerabilities. Many other vulnerability types
require dedicated run-time environments, complex interactions, or
system-level configuration to evaluate, which cannot be easily
incorporated through prompting alone and would require substantial
additional infrastructure. This makes extending the benchmark to
broader vulnerability categories non-trivial and raises questions
about its scalability. Besides, given these constraints, the claims
about comprehensively evaluating AI agents’ cybersecurity capabilities
appear overstated and the scope of the benchmark should be clarified.

**Strengths And Weaknesses:**

Strengths:

1. The paper presents a new benchmark for evaluating AI agents on specific
cybersecurity tasks. Compared with previous benchmarks, CyberCycle includes
the full vulnerability lifecycle, including vulnerability discovery,
PoC generation, and patch generation. This better reflects realistic
bug-fixing workflows compared to prior benchmarks that focus on only one
stage.

2. The paper creates a dataset of 615 real-world vulnerabilities across
120 projects. The construction pipeline can also be
reused to incorporate newly published vulnerabilities.

3. The work further evaluates several recent AI agent frameworks and
models on the benchmark and provides insights into the
current capabilities and limitations of AI agents in bug-finding and bug-fixing tasks.


Weaknesses:

1. The benchmark relies on OSS-Fuzz as the primary source of
vulnerabilities, which limits the variety of programs represented in
the dataset. OSS-Fuzz mostly targets open-source projects that can be
compiled and executed within its fuzzing infrastructure and that
process structured inputs such as files or byte streams. The prompts
(in Appendix A, figure 5) reflect this design, as they instruct the
agent to read and write PoC bytes and the infrastructure simply
executes the program with this input to test whether a crash
occurs. While this does not necessarily indicate a flaw in the
benchmark itself, the paper should explicitly acknowledge that the
diversity of programs is constrained by the characteristics of
OSS-Fuzz.

2. Though the work claims to be scalable, the current benchmark mainly
focuses on memory-safety issues that lead to program crashes. There
are many other types of vulnerabilities, e.g., logic bugs,
authorization issues, injection vulnerabilities, and concurrency bugs,
that do not necessarily trigger crashes and often require more complex
infrastructure to evaluate. For this reason too, the current scope of the
benchmark is limited and the claim of scalability overstated.

3. The evaluation may suffer from artifacts related to
memorization. The benchmark is constructed from historical
vulnerabilities and public repositories. The LLM models used in the
evaluation may have knowledge cutoff dates after the disclosure of
these vulnerabilities or after the release of the corresponding
patches. Hence, the reported performance may partially reflect
memorization instead of reasoning ability. This issue is not
sufficiently discussed.

4. The manual exclusion of dataset instances is described vaguely and
the selection process may have introduced bias. For example, projects
may be excluded due to insufficient test coverage, failing test cases,
or the agent being unable to build the project. However, these issues
are common in real-world software projects, and their presence may be
correlated to the overall quality and complexity of the code
base. Poor test coverage, failing tests, and build issues may indicate
lower quality or more complicated code, which could directly affect
the difficulty of finding and fixing bugs.  Excluding such cases may
bias the benchmark toward cleaner and easier projects, which would
cause evaluations with the benchmark to not reflect the actual
capabilities of AI agents in discovering bugs in the wild.

5. Beyond points 1 and 2, the paper should avoid claiming that the
benchmark can comprehensively evaluate AI agents’ capabilities in
cybersecurity: "cybersecurity" includes many topics not related to bug
finding and bug fixing, which is what this benchmark is about.

---

> ### Author Rebuttal · Authors · 2026-03-31
>
> We thank the reviewer for the constructive feedback.
>
> **Task diversity and scalability (W1,W2,Q1):**
> To make the benchmark representative, we included projects spanning diverse application domains: image/media processing (imagemagick, libraw, libheif), networking (curl, mosquitto, opensips), databases (duckdb), compression (zstd, upx), and cryptography (openssl, boringssl). Many of these are highly popular and complex projects with thousands of GitHub stars. Our automated pipeline is continually running, and the dataset has grown to 910 tasks across 138 projects since submission, demonstrating the pipeline's scalability in practice.
>
> The current evaluation oracle uses sanitizer-triggered crashes, but the evaluation framework is oracle-agnostic: any harness that can programmatically judge success or failure can be integrated. We chose OSS-Fuzz because it provides structured vulnerability data at scale, which is exactly the input our automated pipeline requires. To demonstrate the flexibility and potential of our pipeline, we successfully constructed a task with our pipeline for an uncaught exception vulnerability (CVE-2023-2251) during the rebuttal period.
>
> **Scope concerns (W5):**
> Following the reviewer’s suggestion, we will add a dedicated Limitations and Scope section that explicitly discusses (1) the dataset's current focus on memory-safety vulnerabilities, (2) the characteristics inherited from OSS-Fuzz as the primary data source, and (3) the gap between this scope and "cybersecurity" broadly. We will also note these limitations in other sections where we discuss the scope of the benchmark.
>
> **Memorization concerns (W3):**
> We have conducted a contamination analysis by stratifying end-to-end S3 performance based on whether each vulnerability was disclosed before or after the model's knowledge cutoff date.
>
> |Model|Cutoff|Pre-cutoff|Post-cutoff|Fisher p|Z-test p|
> |-|-|-|-|-|-|
> |Claude Opus 4.5 ($10 cap)|May 2025|19.4%|18.8%|1.00|0.91|
> |Claude Opus 4.5 (no cap)|May 2025|34.2%|34.8%|0.89|0.92|
> |GPT-5.2-Codex ($10 cap)|Aug 2025|21.7%|13.0%|0.19|0.16|
>
> As we can see from the table, both Fisher and Z-test p-values are larger than 0.1, showing that pre-cutoff and post-cutoff performance is statistically indistinguishable. This is consistent with findings from other cybersecurity benchmarks (e.g., Cybench, BountyBench, CyberGym), which have similarly reported that memorization is not a significant factor in agent performance. We were unable to include Gemini 3 Pro in this analysis as the model was deprecated at the time of rebuttal.
>
> **Pipeline filtering statistics (W4/Q2):**
> We provide the filtering criteria and statistics following the stages illustrated in Figure 1:
>
> Stage 1: This stage filtered out approximately half of the data, leaving about 1,400 candidates. Vulnerabilities were excluded if the patch commit message was uninformative or spanned unrelated issues.
>
> Stage 2: About 15% of the data failed this stage (e.g., could not find the vulnerable commit, PoC did not behave as expected). About 1,200 candidates remained.
>
> Stage 3: After agent-assisted test identification, about 800 candidates remained. Failures at this stage were primarily due to the agent being unable to resolve compilation issues, failing to identify tests, or making excessively invasive changes to build scripts.
>
> Stage 4: 74% passed, and 26% were rejected, yielding the final 615 tasks. At this stage, the human expert reviewed each entry for: (1) obscure or redundant code and comments, (2) excessive on-the-fly code changes that break idempotency, (3) redundant dependencies, (4) base image, (5) project metadata, and (6) unit tests.
>
> Regarding the concern about bias toward projects, we acknowledge that excluding projects with failing tests or build issues may skew the dataset toward better-maintained codebases. The retained projects still span diverse application domains (image processing, networking, databases, compilers, compression, cryptography) and are still large and complex (median: 1,769 files, 601K lines of code), with patches ranging from single-line fixes to coordinated edits across 29 files and 1,331 lines (Table 2).
>
> **Insights on how to improve vulnerability discovery (Q3):**
> Our evaluation reveals several actionable insights for agent framework design in Section 4.
> We attached some representative examples here:
>
> Targeted file access matters: Agents using grep/ripgrep to locate relevant snippets substantially outperform those reading entire files into context, which exhausts the context window prematurely.
>
> Successful agents follow a consistent five-phase pattern: keyword extraction, targeted search, code path analysis, initial PoC construction, and iterative refinement via validation feedback.
>
> Domain expertise is a key bottleneck: Analysis failures due to domain-specific knowledge gaps (e.g., CPU emulation, binary parsing, cryptography) are common, suggesting that specialized sub-agents or domain-specific tools could help.

---

> > ### Author Rebuttal · Reviewer_kzxh · 2026-04-03
> >
> > Thank you for the response. The rebuttal addresses my concerns sufficiently that I will raise my score to weak accept, on the assumption that the new experiments and proposed changes to the paper will be incorporated. Various limitations still exist, but they aren't a barrier to acceptance.

---

> > > ### Author Response · Authors · 2026-04-05
> > >
> > > We thank the reviewer for the feedback and for raising the score. We will incorporate all proposed changes and experiments in the paper.

---

### Review · Ethics_Reviewer_s6RH · 2026-03-25

**Recommendation:** Remediation action needed

**Ethics Issue:**

The paper materially advances agentic capability for vulnerability discovery and exploit generation. This is inherently dual-use and could lead to a lowering of the barrier to offensive cyber activity if misapplied. While this dual-use problem is common in security research, it requires a more careful discussion than this paper covers in it's current form.

**Remediation Action:**

I suggest the authors explicitly discuss the dual use problem in the paper. This could be done, for example, in the Impact statement which currently is very limited. This discussion should also cover whether the security problems included in the benchmark were already publicly disclosed and anything related to the specific ethical implications of inclusion of any particular data point in the benchmark.

---

### Decision · Program_Chairs · 2026-04-30

**Decision:**

Accept (regular)

**Comment:**

The authors propose a benchmark for AIs devised to discover and remediate software vulnerabilities. The main strengths of the proposed benchmark are that: (i) it includes the full vulnerability lifecycle; (ii) its construction is semi-automated as it recovers information from real-world historical vulnerabilities; (iii) it provides insight regarding the current capabilities of AIs in bug finding and fixing, and the authors have identified actionable insight to improve vulnerability discovery.
The main concerns of the reviewer regard the scalability, which was overclaimed.
However, the authors promised to discuss in the limitations that the proposed benchmark:
-	Focuses only on memory-safety issues; \
-	Mostly includes open-source projects that can process structured inputs such as files or byte streams.
The other serious concern was that the evaluation may be affected by memorization, which was solved during the rebuttal phase.
One reviewer considered the novelty limited. However, I considered the insights regarding how to improve vulnerability discovery as the novelty in this paper, and I consider them valuable.
Therefore, I recommend acceptance, assuming that the authors will modify the paper as promised in the rebuttal.
I also strongly recommend that they add the summarization of the actionable insight to improve vulnerability discovery in the manuscript.